# A Computer-Aided Approach to Canine Hip Dysplasia Assessment: Measuring Femoral Head–Acetabulum Distance with Deep Learning

Pedro Franco-Gonçalo [1,2,3], Pedro Leite [4], Sofia Alves-Pimenta [2,3,5], Bruno Colaço [2,3,5], Lio Gonçalves [6,7,8], Vítor Filipe [6,7,8], Fintan McEvoy [9], Manuel Ferreira [4] and Mário Ginja [1,2,3,*]

1 Department of Veterinary Science, University of Trás-os-Montes and Alto Douro (UTAD), 5000-801 Vila Real, Portugal; pedrofranco@utad.pt
2 Veterinary and Animal Science Research Centre (CECAV), University of Trás-os-Montes and Alto Douro (UTAD), 5000-801 Vila Real, Portugal; salves@utad.pt (S.A.-P.); bcolaco@utad.pt (B.C.)
3 Associate Laboratory for Animal and Veterinary Sciences (AL4AnimalS), 5000-801 Vila Real, Portugal
4 Neadvance Machine Vision SA, 4705-002 Sequeira, Portugal; pleite@neadvance.com (P.L.); mferreira@neadvance.com (M.F.)
5 Department of Animal Science, University of Trás-os-Montes and Alto Douro (UTAD), 5000-801 Vila Real, Portugal
6 School of Science and Technology, University of Trás-os-Montes and Alto Douro (UTAD), 5000-801 Vila Real, Portugal; lgoncalv@utad.pt (L.G.); vfilipe@utad.pt (V.F.)
7 Department of Engineering, University of Trás-os-Montes and Alto Douro (UTAD), 5000-801 Vila Real, Portugal
8 Institute for Systems and Computer Engineering (INESC-TEC), Technology and Science, 4200-465 Porto, Portugal
9 Department of Veterinary Clinical Sciences, Faculty of Health and Medical Sciences, University of Copenhagen, 1165 Copenhagen, Denmark; fme@sund.ku.dk
* Correspondence: mginja@utad.pt

**Abstract:** Canine hip dysplasia (CHD) screening relies on radiographic assessment, but traditional scoring methods often lack consistency due to inter-rater variability. This study presents an AI-driven system for automated measurement of the femoral head center to dorsal acetabular edge (FHC/DAE) distance, a key metric in CHD evaluation. Unlike most AI models that directly classify CHD severity using convolutional neural networks, this system provides an interpretable, measurement-based output to support a more transparent evaluation. The system combines a keypoint regression model for femoral head center localization with a U-Net-based segmentation model for acetabular edge delineation. It was trained on 7967 images for hip joint detection, 571 for keypoints, and 624 for acetabulum segmentation, all from ventrodorsal hip-extended radiographs. On a test set of 70 images, the keypoint model achieved high precision (Euclidean Distance = 0.055 mm; Mean Absolute Error = 0.0034 mm; Mean Squared Error = $2.52 \times 10^{-5}$ mm$^2$), while the segmentation model showed strong performance (Dice Score = 0.96; Intersection over Union = 0.92). Comparison with expert annotations demonstrated strong agreement (Intraclass Correlation Coefficients = 0.97 and 0.93; Weighted Kappa = 0.86 and 0.79; Standard Error of Measurement = 0.92 to 1.34 mm). By automating anatomical landmark detection, the system enhances standardization, reproducibility, and interpretability in CHD radiographic assessment. Its strong alignment with expert evaluations supports its integration into CHD screening workflows for more objective and efficient diagnosis and CHD scoring.

**Keywords:** canine hip dysplasia; screening; artificial intelligence; computer vision; regression network; semantic segmentation

## 1. Introduction

Canine Hip Dysplasia (CHD) is a hereditary orthopedic disorder that predominantly affects large and rapidly growing dog breeds. It is characterized by abnormal development of the hip joint, resulting in laxity (looseness) of the joint, where the femoral head (the rounded top part of the thigh bone) does not fit securely within the acetabulum (hip socket), leading to instability [1]. Over time, this instability promotes abnormal joint wear, which often progresses to degenerative joint disease (DJD), commonly known as osteoarthritis, a condition where the joint becomes inflamed, painful, and stiff due to the breakdown of cartilage and the development of bone spurs. In milder cases, instability causes subluxation (partial dislocation of the femoral head from the acetabulum), while in more severe cases, it can progress to luxation (complete dislocation) [1]. The development of CHD is influenced by a complex interaction of genetic predisposition and environmental factors, such as diet and exercise. Radiographic imaging remains the cornerstone of CHD diagnosis, with the ventrodorsal hip-extended (VDHE) view serving as the standard imaging technique for evaluating hip joint conformation, in which the dog lies on its back with its hind legs stretched out to provide a clear view of the hip joints. This approach is vital for guiding clinical management and informing breeding practices to reduce CHD prevalence [1].

The screening process for CHD follows standardized radiographic scoring systems developed by major international organizations such as the Fédération Cynologique Internationale (FCI), British Veterinary Association/Kennel Club (BVA/KC), and Orthopedic Foundation for Animals (OFA). The FCI classifies hips into five grades (A–E) and has historically relied on the Norberg Angle (NA) to assess joint subluxation and acetabular morphology. The BVA/KC system evaluates multiple morphological parameters, incorporating the NA, subluxation, and DJD criteria. In contrast, the OFA applies a categorical grading based on hip joint congruency without numerical scoring [1,2]. Building upon these methodologies, Mark Flückiger introduced a refined scoring system in 1993 to enhance CHD assessment objectivity while aligning with FCI standards. This system incorporated six radiographic parameters, prominently including the NA and the position of the femoral head center relative to the dorsal acetabular edge (FHC/DAE), alongside signs of DJD, to provide a holistic view of hip joint health [3,4].

The NA has traditionally been used as a key metric for CHD scoring according to FCI criteria, assessing the angular positioning of the femoral head relative to the acetabulum, thereby reflecting the extent of subluxation and acetabular depth [5]. An NA of $\geq 105°$ is typically viewed as indicative of sufficient hip joint integrity, implying a lower risk of developing DJD. However, studies have revealed limitations in using the NA alone to predict DJD susceptibility [6–8]. Culp et al. demonstrated that an NA of $\geq 105°$ may fail to identify dogs at risk for DJD, as it can overlook joint laxity, a critical risk factor that becomes apparent only under stress or outside the standard VDHE radiographic view [6]. Consequently, some dogs with an $NA \geq 105°$ may have underlying hip laxity, predisposing them to diagnostic inaccuracies and resulting in false negatives, allowing dysplastic genes to persist in breeding populations. Additionally, NA thresholds can vary by breed; for instance, Australian Shepherds and Borzois may exhibit NA values $< 105°$ due to unique pelvic structures rather than true joint laxity [6,9,10].

Further complicating the issue, studies have demonstrated significant variability in inter-rater agreement for NA measurements. Verhoeven et al. observed an average inter-rater agreement rate of 72% when distinguishing between "normal" and "dysplastic" hips using the NA [11]. Similarly, Geissbühler et al. reported a poor-to-fair level of agreement, with an intraclass correlation coefficient (ICC) of 0.47 and a Cohen's kappa value of 0.29, indicating substantial inconsistency among examiners [12]. The inherent variability observed in these findings, likely stemming from the high level of technical

proficiency required for precise NA measurement, including ambiguities in defining the effective craniolateral acetabular rim (CrAR) [12,13], highlights why the NA should not serve as the primary or most heavily weighted metric within CHD scoring protocols.

The FHC/DAE distance directly evaluates the horizontal positioning of the femoral head relative to the edge of the acetabulum, indicating whether the femoral head is sufficiently covered by the acetabulum or if it is positioned outward [4], which would signal joint laxity and potential instability. Even though it seems to share functional overlap with the NA, this redundancy can help address the limitations of the NA, especially in cases where it may not detect dogs at risk for DJD. While it may not have the same widespread recognition as the NA, Skurková et al. showed that excluding FHC/DAE from the Flückiger system resulted in a 21.24% increase in false negatives, indicating that more cases of CHD were under-diagnosed when FHC/DAE was omitted. By contrast, omitting NA increased false negatives by only 11.06%, underscoring the critical role of FHC/DAE in capturing lateral femoral head displacement, which NA alone might overlook. Furthermore, the study found that concordance with the original Flückiger system was higher when NA was omitted (84.96%) compared to FHC/DAE (78.32%) [7]. This suggests that FHC/DAE provides unique insights into hip joint conformation, adding specificity to CHD diagnosis that NA cannot fully replace.

Advances in technology have enabled the development of computer-aided detection (CAD) systems for medical imaging, significantly enhancing diagnostic precision [14]. While early CAD systems often struggled with low precision, deep learning (DL) algorithms have demonstrated remarkable improvements, achieving near-human performance in many imaging tasks [14]. This progress has spurred interest in using DL techniques to address diagnostic challenges in veterinary medicine. McEvoy et al. initially utilized a partial least squares discriminant analysis model and a nonlinear neural network to classify VDHE radiographs as either "hip" or "not hip", resulting in classification errors of 6.7% and 8.9%, respectively [15]. Building upon this foundation, they later developed a two-step method using a YoloV3 Tiny, a convolutional neural network (CNN), to detect and isolate regions of interest for each hip joint, achieving an intersection over union (IoU) score of 0.85, indicating strong agreement with ground truth data. In the second step, another YoloV3 Tiny CNN was trained for binary classification of CHD (i.e., distinguishing healthy from dysplastic hips). Despite reducing computational complexity by focusing on the regions of interest, this approach achieved a sensitivity of only 53% for identifying dysplastic cases. Subsequently, they applied a YoloV3 Tiny CNN to detect and categorize hip joints using FCI criteria, achieving a sensitivity of 53% and a specificity of 92% [16]. Similarly, Gomes et al. applied a pre-trained Inception-V3 model for binary classification, attaining 83% sensitivity for dysplastic cases but encountering specificity challenges, with a rate of 66% due to false positives [17]. Wang et al. used a DL model known as EfficientNet to classify CHD in radiographic images, achieving an area under the receiver operating characteristic curve (AUC) of 0.964 and 89.1% accuracy in binary classification, and an AUC of 0.913 for FCI grading [18]. Conversely, Akula et al. explored a different approach, utilizing a 3D CNN trained on magnetic resonance imaging scans, attaining 89.7% accuracy in binary classification and highlighting the potential of volumetric analysis for comprehensive CHD assessment [19]. While convolutional layers are well suited for extracting visual features from images and fully connected layers typically handle the final classification task, the "black-box" nature of these models complicates interpretation and validation of predictions, potentially undermining clinical trust and reliability [20]. Therefore, there is a pressing need for more interpretable approaches that clarify model reasoning and enhance prediction transparency. In our view, incorporating objective numeric feature analysis into CHD scoring systems is crucial for precise and reliable assessment of hip joint conformation.

Recent research has focused on applying DL systems to precisely measure key objective metrics such as the Hip Congruency Index (HCI) [21,22] and Femoral Neck Thickness Index (FNTi) [23,24], both valuable for assessing hip joint health. A multidisciplinary team developed a DL-based approach using transfer learning to segment and delineate the femoral head and acetabulum from VDHE images, enabling accurate HCI calculations to quantify how well the femoral head fits within the acetabulum. Built on a U-Net architecture with EfficientNet modules, the model achieved high segmentation accuracy, with Dice scores (DS) of 0.98 for the femur and 0.93 for the acetabulum. This strong performance closely matched manual expert evaluations, demonstrating potential to reduce subjectivity and improve consistency in CHD assessments [22]. Building on this framework, researchers applied DL to measure the FNTi, an indicator of biomechanical stress and adaptation in dysplastic hips. The system combined a YOLOv4 network for joint detection with a ResNetv2 backbone for anatomical landmark prediction. Integrated with the segmentation framework used for HCI, this approach enabled precise extraction of femur border coordinates and femoral head circumferences to compute the FNTi, achieving an IoU score of 0.96 [24]. The resulting index showed a significant correlation with CHD severity according to FCI grading [23], and automated FNTi measurements demonstrated strong agreement with expert manual evaluations, with an ICC of 0.88, underscoring its reliability [23,24]. Together, these automated systems for HCI and FNTi highlight the transformative potential of DL-driven approaches to enhance objectivity, reliability, and transparency in CHD screening, paving the way for more precise and reliable assessments within veterinary orthopedics. As another step in this direction, this study introduces a system for automated measurement of the FHC/DAE distance. Unlike previous AI models focused on direct CHD classification, this approach emphasizes transparency by providing a quantifiable feature to support multimetric CHD radiographic assessment.

The primary objectives of this study were to develop the FHC/DAE system and evaluate its agreement and reliability with experienced examiners (P.F-G. and M.G.) to validate its clinical applicability. Our null hypothesis is that there is no agreement beyond chance in the FHC/DAE classifications made by the two examiners and between the examiners and the automated system.

## 2. Materials and Methods

### 2.1. Study Design

This study was conducted in two distinct parts, each addressing complementary objectives to develop and validate a DL system designed to assist in the radiographic evaluation of CHD.

The first part focused on the development of the DL system, which integrates two key models: a keypoint model to detect the FHC and a segmentation model to delineate the DAE. Together, these models enable precise calculation of the FHC/DAE metric, a key parameter in assessing hip joint conformation (Figure 1).

The second part of the study aimed to validate the DL system under practical conditions by comparing its performance against those of experienced examiners.

All images included in this study were in Digital Imaging and Communications in Medicine (DICOM) format and met the required criteria for CHD screening. These criteria specified that dogs had to be over 12 months of age and radiographs needed to demonstrate clear reference landmarks and adequate technical quality for CHD screening. This included proper positioning on the X-ray table and image clarity, ensuring the DAE was sufficiently visible and unobscured. For cases where exact age information was unavailable, skeletal maturity was used as a validation criterion, as only dogs over 12 months are eligible

for CHD screening. Due to the observational nature of the study, the ethical committee approval and the owner's consent were waived.

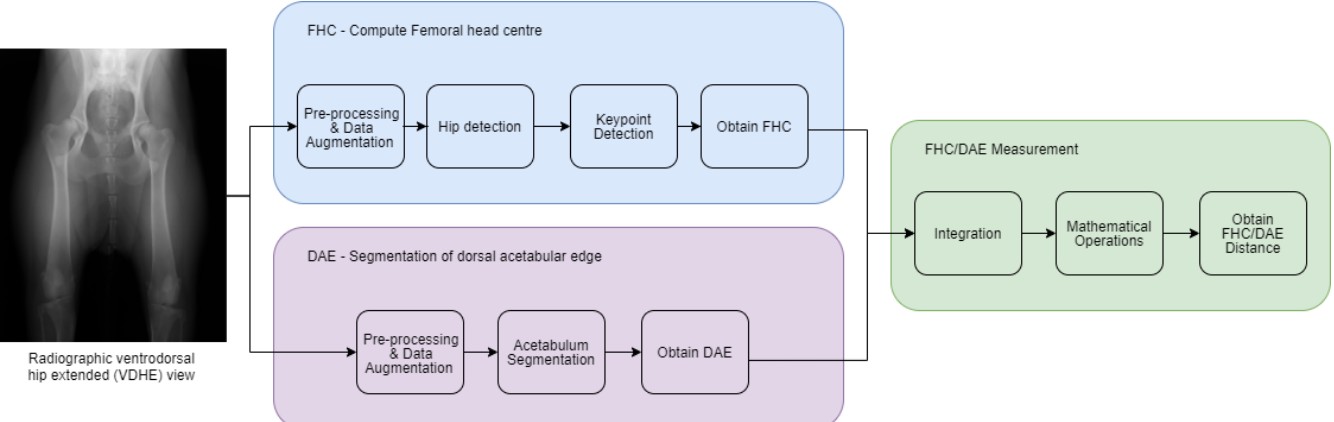

**Figure 1.** Overview of the system pipeline for FHC/DAE measurement. The process initiates with the input of a ventrodorsal hip-extended (VDHE) radiographic image, followed by parallel computational models for detecting the femoral head center (FHC) and the dorsal acetabular edge (DAE) using segmentation and keypoint detection models. The extracted coordinates are integrated into mathematical operations to calculate the FHC/DAE distance.

### 2.2. First Part: Automated System Overview

The objective of the automated FHC/DAE measurement system was to measure the distance between the FHC and the DAE from VDHE radiographic images. To achieve this, the development of the system was structured into three stages: FHC detection, DAE segmentation, and FHC to DAE distance calculation.

### 2.2.1. Stage 1: FHC Detection

The first phase of the proposed methodology focused on the detection of the FHC. For this purpose, two different datasets were used to train the regression model: one containing the spatial location of the hip joints and the other comprising specific landmark coordinates around the femoral head.

For the first dataset, 7967 annotated VDHE images from the University of Copenhagen and the Danish Kennel Club databases were used, with bounding boxes indicating the spatial locations of the hip joints (Figure 2).

For the second dataset, 571 VDHE images were obtained from the Veterinary Hospital of the UTAD and the Danish Kennel Club databases. These images had previously been uploaded to the Dys4Vet 25 software as part of an earlier project focused on NA calculation, in which P.F.G. performed manual annotations to train an automated NA model.

Although NA is not the focus of the present study, the annotations generated during that earlier work proved valuable for the development of the current FHC/DAE system. The NA was calculated using a semi-automatic method in Dys4Vet. For each hip joint, four anatomical landmarks were annotated: P1, P2, and P3, positioned around the femoral head to define a circle accurately representing its curvature, and P4, marking the effective CrAR. Using these annotations, the software automatically computed the angle between the line connecting the centers of the generated circles and the line extending from each center to the respective ipsilateral CrAR.

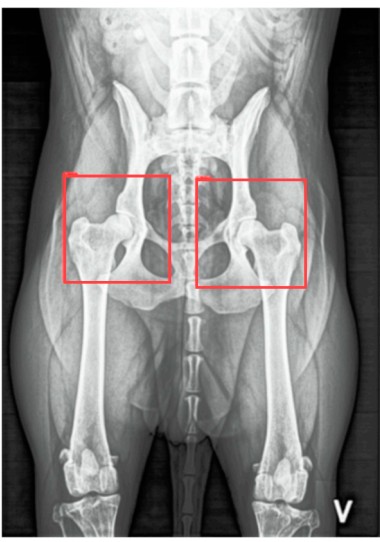

**Figure 2.** Illustration of boundary boxes demarcating the hip joints on a ventrodorsal hip-extended (VDHE) radiographic image, representing the regions of interest.

Importantly, the three femoral head points (P1, P2, P3) were repurposed in the current study as ground truth for training the regression network used to identify the FHC. The image crops from both sides were combined due to the symmetry introduced by applying a flip, resulting in identical outputs.

Model Training

The training process for the model began with a hip detection experiment utilizing the YOLOv4 architecture. This step aimed to locate the hip joints within the VDHE radiographic images accurately. The dataset of 7967 images was divided into three subsets: 70% allocated for training, 15% reserved for validation to save checkpoints and verify early stopping, and 15% for testing. The training utilized images resized to 416 × 416 pixels, processed over 300 iterations with a learning rate of 0.001 and a batch size of 4. Data augmentation was applied to improve the model's generalizability and reduce the risk of overfitting. By introducing controlled variability into the training data, augmentation helps simulate real-world differences that may arise from variations in imaging equipment, patient positioning, or acquisition settings. Specifically, transformations involving zoom, horizontal flip, perspective adjustments, and brightness alterations were applied, each with a probability of 0.5 of being executed. These operations not only enhance robustness to minor geometric and photometric variations but also support effective training under limited data conditions, reducing dependence on large annotated datasets.

Upon completing the training, the YOLOv4 model was applied to the dataset of 571 images, producing 1142 cropped images resized to 224 × 224 pixels, each representing one of the two hip joints. The cropped images were then used in the keypoints regression experiment to identify the specific anatomical landmarks for FHC detection. The dataset was partitioned, with 80% allocated for training, 10% reserved for validation, and 10% for testing. The ResNet50v2 architecture, pre-trained on the ImageNet dataset, was selected as the backbone network due to its robust feature extraction capabilities. The architecture was modified to suit the regression task by replacing its final layer with a two-dimensional Global Average Pooling layer, which produced a vector comprising 2048 features. This was followed by two dense layers with 1024 and 512 neurons, respectively, each incorporating a dropout regularization factor of 0.1. Finally, a dense layer with 8 neurons was used to compute the coordinates of the four keypoints: $P1_x$, $P1_y$, $P2_x$, $P2_y$, $P3_x$, $P3_y$, $P4_x$, $P4_y$.

The regression model was trained for a maximum of 500 epochs with early stopping, utilizing an initial learning rate of 0.001 and a batch size of 8. If no improvement was observed over 25 consecutive epochs, the learning rate was halved to facilitate convergence. The Adam optimizer was employed to update the network's parameters, and the Euclidean distance was used as the loss function. Data augmentation techniques were applied during training to improve the model's robustness and generalization. These included horizontal flipping, shear transformation ($-15°$ to $15°$), rotation ($-5°$ to $5°$), scaling (0.75 to 1.15), translation ($-0.05$ to 0.05), perspective adjustment (0.02 to 0.05), and Gaussian blur (0 to 0.5). Each transformation was applied with a probability of 0.5.

After the training phase, the FHC for each hip joint was determined by calculating the centroid of the three keypoints (P1, P2, P3) predicted by the model (Figure 3).

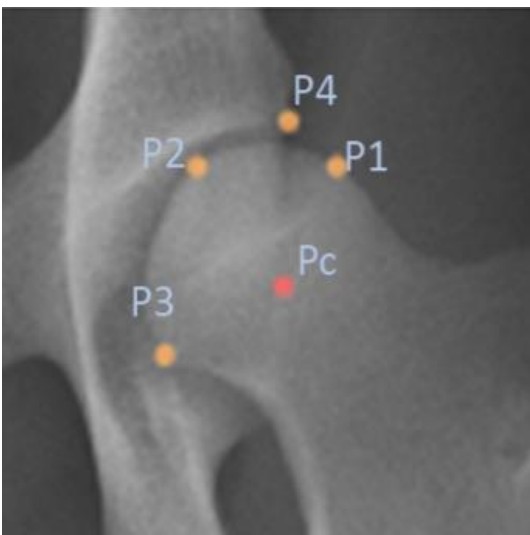

**Figure 3.** Visualization of the keypoints (P1, P2, P3, P4) predicted on the femoral head and acetabulum. The centroid (Pc), calculated from P1, P2, and P3, represents the femoral head center (FHC).

Model Performance

The performance of the FHC detection model was evaluated using multiple metrics to assess its accuracy and precision, including Euclidean Distance (ED), Mean Absolute Error (MAE), and Mean Squared Error (MSE). The ED measures the straight-line distance between the predicted and actual coordinates of the FHC, while MAE (Equation (1)) and MSE provide statistical measures of error magnitude, with MSE placing greater emphasis on larger errors due to its squared component (Equation (2)).

$$MAE = \frac{1}{n}\sum_{i=1}^{n}|x_i - y_i| \tag{1}$$

$$MSE = \frac{1}{n}\sum_{i=1}^{n}(x_i - y_i)^2 \tag{2}$$

where $x_i$ is the ground truth value, $y_i$ is the prediction value, and $n$ is the total number of points.

### 2.2.2. Stage 2: DAE Boundary Detection

The second stage of the proposed methodology focused on detecting the DAE using a semantic segmentation network for bone identification. This process relied on a single dataset comprising 624 VDHE images obtained from the Veterinary Hospital of UTAD and the Danish Kennel Club databases, which was used to train the segmentation model.

To prepare the dataset for training, each VDHE image was manually segmented using the LabelMe polygonal image annotation tool [25]. P.F.-G. carefully delineated and labeled the femur and acetabulum in both hip joints, ensuring accurate and well-defined anatomical boundaries (Figure 4).

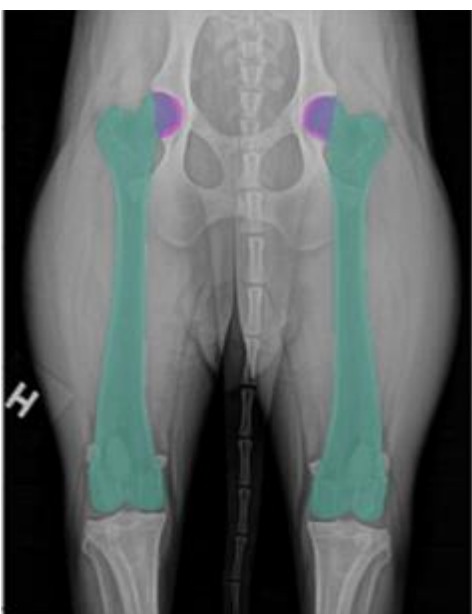

**Figure 4.** Anatomical annotations of the femur and acetabulum. The green regions correspond to the femur, the pink regions represent the acetabulum, and the blue areas indicate the overlapping regions between the femoral head and the acetabulum.

The segmentation model utilized in this study builds upon our previous works [22,24] and employs a U-Net architecture with EfficientNet modules as the feature extraction backbone. In this configuration, the EfficientNetB4 component, pre-trained on the ImageNet dataset, functions as the encoder, capturing complex features from the input, while the U-Net serves as the decoder, transforming these features into a detailed segmentation mask.

Following the annotation process, the segmented images were converted into three-channel binary masks, where each channel represented one of the following classes: (1) background (non-femur and non-acetabulum regions of the DICOM image), (2) femur, and (3) acetabulum. The radiographs were resized to $448 \times 448$ pixels, and the corresponding masks were adjusted using nearest-neighbor interpolation to preserve their binary nature.

Model Training

To train the segmentation model, the dataset of 624 images was split into three subsets: 80% for training, 10% for validation, and 10% for testing. The training process employed the Adam optimizer with a learning rate of 0.001 and a batch size of 8, leveraging two GPUs to accelerate computation. To enhance the dataset's diversity and improve model generalization, extensive data augmentation techniques were applied, including horizontal flipping, shifting ($-0.06$ to $0.06$), rotation ($-5°$ to $5°$), scaling (0.9 to 1.1), and adjustments to contrast and brightness ($-0.2$ to $0.2$). Each transformation was executed with a probability of 0.5. To address class imbalance within the data, a hybrid loss function was implemented, combining Dice loss and focal loss. This approach ensured the model accounted for both easily segmented regions and challenging regions with lower pixel representation, improving segmentation performance.

The DAE was not explicitly segmented or highlighted as a separate structure. Instead, it was determined as the closest point on the acetabular contour to the FHC. By calculating

the shortest horizontal distance between the FHC and the acetabulum, the location of the DAE was effectively incorporated into the measurement process.

Model Performance

The performance of the DAE segmentation model was evaluated based on its ability to accurately delineate the acetabulum, the primary structure of interest in this study. Model accuracy was assessed using two key metrics: the DS and the IoU. The DS measures the similarity between the predicted segmentation masks and the ground truth annotations, indicating the degree of overlap (Equation (3)). The IoU quantifies the proportion of the intersection between the predicted and ground truth areas relative to their total combined area, providing a comprehensive measure of segmentation performance (Equation (4)).

$$Dice = \frac{2 \times |X \cap Y|}{|X| + |Y|} \tag{3}$$

$$IoU = \frac{|X \cap Y|}{|X \cup Y|} \tag{4}$$

where $X$ is the ground truth mask and $Y$ is the predicted segmentation mask.

### 2.2.3. Stage 3: Computing FHC/DAE Measurement

The task of determining the distance between the FHC and the DAE can be expressed as a mathematical problem. The FHC is represented as a center point, $(x_c, y_c)$, while the acetabular contour is defined as a set of points, $P = \{(x_1, y_1), (x_2, y_2), \ldots, (x_n, y_n)\}$, where each point $(x_i, y_i)$ corresponds to a vertex on the acetabular contour, $C$.

To calculate the distance, the algorithm identifies the intersection of a horizontal line passing through the center point $(x_c, y_c)$ with the acetabular contour, $C$. This is achieved by iterating through all consecutive pairs of contour points $(x_i, y_i)$ and $(x_{i+1}, y_{i+1})$ and determining whether $y = y_c$ intersects the segment connecting the points $(x_i, y_i)$ and $(x_{i+1}, y_{i+1})$ ($x_{\text{intersect}}$). For any segment where this condition is satisfied, the x-coordinate of the intersection point ($x_{\text{intersect}}$) is calculated using linear interpolation (see illustration of the procedure bellow in Figure 5):

$$x_{\text{intersect}} = x_i + \frac{y_c - y_i}{m} \tag{5}$$

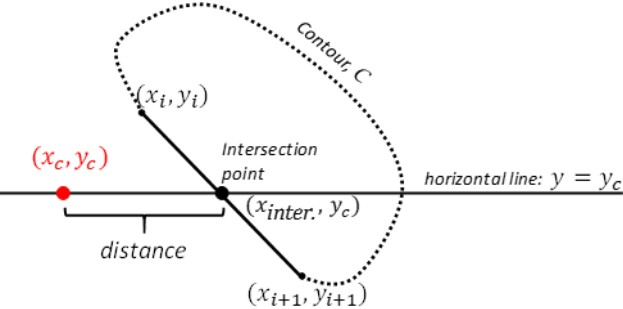

**Figure 5.** Illustration of the procedure to compute the distance between the femoral head center (FHC) and the dorsal acetabular edge (DAE).

The slope of the line segment is then computed as:

$$m = \frac{y_{i+1} - y_i}{x_{i+1} - x_i} \tag{6}$$

Finally, the signed distance between the FHC and the DAE is determined using the following equation:

$$Signed_{\text{distance}}(mm) = side \cdot (x_{\text{intersect}} - x_c) \cdot pixelspacing_x \tag{7}$$

where the parameter *side* indicates the direction of the distance relative to the FHC (Figure 6).

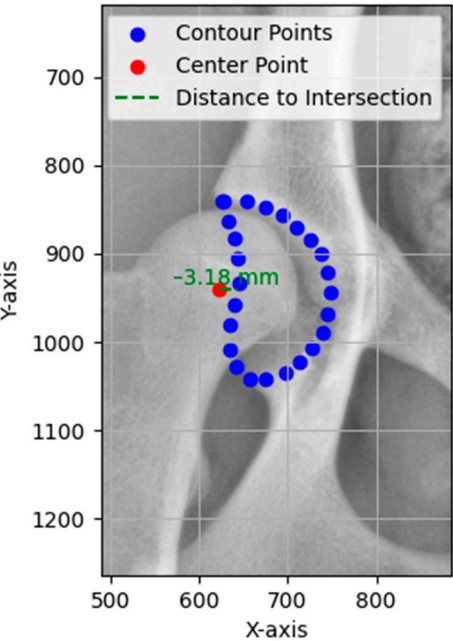

**Figure 6.** Illustration of the application of the algorithm for calculating the horizontal distance between the femoral head center (FHC) and the nearest intersection point on the acetabular contour on a VDHE radiographic image. Each plot demonstrates the identified contour points (blue), the calculated center point (red), and the horizontal distance ($-3.18$ mm) to the intersection (green dashed line).

### 2.3. Second Part: Comparative Analysis for the Automated System Validation

This second part of the study focused on validating the automated FHC/DAE system through a comparative analysis with expert examiners. A total of 70 VDHE radiographs from dogs (140 hip joints) were selected from the Veterinary Hospital of UTAD and the Danish Kennel Club databases. These radiographs were acquired for CHD screening between 2010 and 2024, and each dog's age, breed, and sex were recorded. To ensure a diverse and representative sample covering all degrees of CHD, images for the second part of the study were selected based on the NA values of the hip joints. From a previously annotated dataset of 1150 VDHE radiographs, approximately equal numbers of images were chosen for five predefined NA categories: values of 105° or more, around 105°, around 100°, around or greater than 90°, and below 90°. This approach yielded a balanced dataset reflecting a broad spectrum of hip joint conformations.

In this part of the study, FHC/DAE measurements were performed on the selected 140 hip joints using DICOM format images. The analyses were conducted independently by two experienced examiners, P.F.-G. (Examiner 1, E1) and M.G. (Examiner 2, E2). Following the training phase, the system was used to generate FHC/DAE measurements for the same 140 hip joints. Measurements obtained from the two examiners and the system were then compared to assess reproducibility, both between examiners and between examiners and the system.

The FHC/DAE metric reflects the positional relationship (in millimeters) between the FHC and the DAE. According to Flückiger's CHD point-system protocol, positive FHC/DAE values indicate that the FHC is positioned medially relative to the DAE, while negative values denote a lateral position. In clinical terms, negative values correspond to reduced acetabular coverage, whereas positive values suggest improved coverage and joint congruity. A value of 0 signifies perfect superimposition between the FHC and DAE. For the purposes of this study, and to ensure clinical relevance and comparability, FHC/DAE measurements in this study were categorized according to Flückiger's protocol into six relationship-based groups: medial+ (>2 mm), medial (≤2 to 1 mm), superimposed (<1 to −1 mm), lateral (<−1 to −5 mm), lateral+ (−6 to −10 mm), and luxation (<−10 mm) [3]. Values between the lateral and lateral+ categories were rounded to the nearest integer.

As performed by the examiners, the FHC/DAE metric was manually determined through a two-step process using the Dys4Vet image analysis software. First, the FHC was determined by drawing a circle encompassing the femoral head to locate its center. Then, a horizontal line, roughly parallel to the image width, was drawn to measure the distance between the FCH and the DAE (Figure 7).

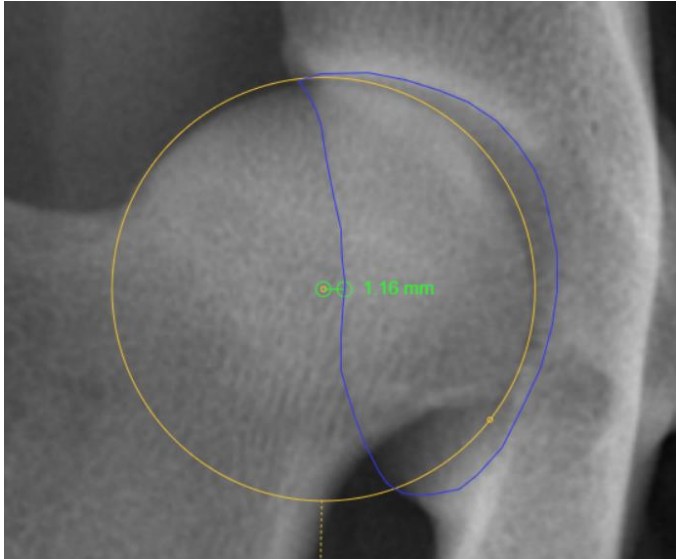

**Figure 7.** Manual measurement of the FHC/DAE metric using the Dys4Vet software on a ventrodorsal hip-extended (VDHE) radiographic image. The femoral head center (FHC) was determined by fitting a circle (yellow) around the femoral head while the dorsal acetabular edge (DAE) was identified along the acetabulum contour (blue). The measured distance between the FHC and DAE is indicated by the green line (−1.16 mm; minus sign not shown in the image). The acetabulum was highlighted for illustrative purposes to enhance the visibility of the DAE but this step was not required in practice.

Statistical Analysis

Statistical analysis was performed using the SPSS software (SPSS Statistics for Windows 128 Version 27.0: IBM Corp., Armonk, NY, USA). A *p*-value of <0.05 was considered statistically significant. The Central Limit Theorem was adopted, which stipulates that for sufficiently large sample sizes (*n* > 30), the distribution tends to be normally distributed, regardless of the original distribution of the variable in the population; therefore, parametric tests for data analysis were used [26].

In this study, weighted kappa was used to assess the agreement between FHC/DAE classifications made by the two examiners and between the examiners and the automated FHC/DAE system. While kappa is sometimes described as a reliability metric, particularly when assessing the repeatability of measurements, its primary function in this context is to quantify the level of agreement beyond chance [27,28]. Weighted kappa was selected for its

suitability for ordinal data, as it accounts for chance agreement and assigns penalties to disagreements based on their magnitude. Linear weights were applied to disagreements, assigning proportional penalties based on the distance between categories, ensuring that disagreements between adjacent categories were penalized less than those between distant categories [27]. Agreement values were interpreted as follows: poor (<0), slight (0–0.2), fair (0.2–0.4), moderate (0.4–0.6), substantial (0.6–0.8), and almost perfect (>0.8) [29]. Confidence intervals (95% CI) for weighted kappa were calculated to evaluate the precision of the agreement estimates [28]. Contingency tables (crosstabs) were used to organize and analyze the classifications, providing a basis for the calculation of agreement metrics [27]. Observed agreement (*Po*), calculated as the proportion of classifications that matched exactly, was also reported as a complementary measure of overall concordance between raters [27].

The intraclass correlation coefficient ($ICC_{3,1}$ absolute agreement model) was used to evaluate the inter-rater reliability of quantitative FHC/DAE measurements, alongside the standard error of measurement (SEM) to assess the precision of individual measurements [30–32]. ICC values were interpreted as follows: random (0), poor (<0.5), moderate (0.5–0.75), good (0.75–0.9), excellent (>0.9), and perfect (1) reliability [31]. A lower limit 95% CI of ICC > 0.75 was defined as adequate reliability [31,33].

## 3. Results

### 3.1. Performance of the Models Comprising the Automated System

The overall ED for the FHC detection model was 0.0055, with individual keypoint errors recorded as follows: ED_P1 = 0.0048, ED_P2 = 0.0056, and ED_P3 = 0.0038. On the test set, the model recorded a loss of 0.0134, a MAE of 0.0034 mm, and a MSE of $2.52 \times 10^{-5}$ mm$^2$ (Figure 8).

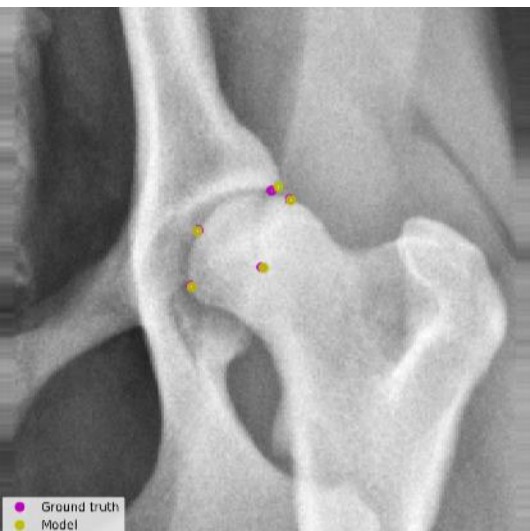

**Figure 8.** Comparison between the ground truth and the predictions made by the model in a test image.

The DAE segmentation model produced a loss of 0.0207, a DS of 0.96, and an IoU score of 0.92.

### 3.2. Comparison Between Human Examiners

The 70 images included in the second part of this study belonged to 70 dogs, predominantly from five main breeds: Portuguese Mountain Dogs (12 dogs, 17.14%), Labrador Retrievers (9 dogs, 12.86%), German Shepherds (6 dogs, 8.57%), Golden Retrievers (6 dogs,

8.57%), Portuguese Pointers (6 dogs, 8.57%), along with several other breeds. There were 48% males and 52% females.

The mean $\pm$ standard deviation (SD) of continuous FHC/DAE measurements for E1 was $-1.32 \pm 2.70$ mm and for E2 was $-1.53 \pm 2.70$ mm. For the comparison between E1 and E2, the weighted kappa was $\kappa = 0.90$ (95% CI [0.84, 0.95], $p < 0.001$), with $Po = 90\%$ (Table 1). The ICC between E1 and E2 was 0.97 (95% CI [0.96, 0.98], $p < 0.001$), with a SEM of 0.91 mm.

**Table 1.** Contingency Table of FHC/DAE Classifications Between Examiner 1 (E1) and Examiner 2 (E2).

| | | E2 | | | | | | |
|---|---|---|---|---|---|---|---|---|
| | | Medial+ | Medial | Super Imposed | Lateral | Lateral+ | Luxation | Total |
| | Medial+ | 6 | 0 | 0 | 0 | 0 | 0 | 6 |
| | Medial | 1 | 7 | 1 | 0 | 0 | 0 | 9 |
| | Super Imposed | 0 | 2 | 59 | 6 | 0 | 0 | 67 |
| E1 | Lateral | 0 | 0 | 1 | 43 | 2 | 0 | 46 |
| | Lateral+ | 0 | 0 | 0 | 1 | 10 | 0 | 11 |
| | Luxation | 0 | 0 | 0 | 0 | 0 | 1 | 1 |
| | Total | 7 | 9 | 61 | 50 | 12 | 1 | 140 |

*3.3. Comparisons Between Human Examiners and the Automated System*

The mean $\pm$ SD of continuous FHC/DAE measurements for the system was $-1.12 \pm 2.59$ mm. For the comparison between E1 and the system, the weighted kappa was $\kappa = 0.86$ (95% CI [0.80, 0.92], $p < 0.001$), with $Po = 87\%$ (Table 2). For the comparison between E2 and the system, the weighted kappa was $\kappa = 0.79$ (95% CI [0.72, 0.86], $p < 0.001$), with $Po = 80\%$ (Table 3). The ICC between E1 and the system was 0.97 (95% CI [0.95, 0.98], $p < 0.001$), with a SEM of 0.92 mm. The ICC between E2 and the system was 0.93 (95% CI [0.88, 0.96], $p < 0.001$), with a SEM of 1.34 mm.

**Table 2.** Contingency Table of FHC/DAE Classifications Between Examiner 1 (E1) and System.

| | | System | | | | | | |
|---|---|---|---|---|---|---|---|---|
| | | Medial+ | Medial | Super Imposed | Lateral | Lateral+ | Luxation | Total |
| | Medial+ | 5 | 1 | 0 | 0 | 0 | 0 | 6 |
| | Medial | 1 | 6 | 2 | 0 | 0 | 0 | 9 |
| | Super Imposed | 0 | 5 | 62 | 0 | 0 | 0 | 67 |
| E1 | Lateral | 0 | 0 | 6 | 39 | 1 | 0 | 46 |
| | Lateral+ | 0 | 0 | 0 | 2 | 9 | 0 | 11 |
| | Luxation | 0 | 0 | 0 | 0 | 0 | 1 | 1 |
| | Total | 6 | 12 | 70 | 41 | 10 | 1 | 140 |

**Table 3.** Contingency Table of FHC/DAE Classifications Between Examiner 2 (E2) and System.

| | | System | | | | | | |
|---|---|---|---|---|---|---|---|---|
| | | Medial+ | Medial | Super Imposed | Lateral | Lateral+ | Luxation | Total |
| E2 | Medial+ | 5 | 2 | 0 | 0 | 0 | 0 | 7 |
| | Medial | 1 | 5 | 3 | 0 | 0 | 0 | 9 |
| | Super Imposed | 0 | 5 | 56 | 0 | 0 | 0 | 61 |
| | Lateral | 0 | 0 | 11 | 37 | 2 | 0 | 50 |
| | Lateral+ | 0 | 0 | 0 | 4 | 8 | 0 | 12 |
| | Luxation | 0 | 0 | 0 | 0 | 0 | 1 | 1 |
| | Total | 6 | 12 | 70 | 41 | 10 | 1 | 140 |

## 4. Discussion

The development of an automated system for FHC/DAE measurement represents a significant step toward improving the objectivity and reliability of CHD assessments. This study evaluated the system's agreement with expert human examiners to determine its clinical applicability, ensuring that automated classifications align with established methodologies. The results demonstrated strong inter-rater agreement and reliability between the two human examiners (E1 and E2), as well as high concordance between the system and E1, who served as the reference standard. These findings allow us to reject the null hypothesis, which stated that there is no agreement beyond chance in the FHC/DAE classifications made by the two examiners and between the examiners and the automated system. Additionally, the system exhibited consistent performance across both frequent and less common classification categories, supporting its potential as a robust tool to assist in CHD scoring. However, some classification tendencies were noted, particularly a slight shift toward less severe categories, which may have implications for clinical decision-making. The system also demonstrated strong performance, with precise keypoint detection by the FHC model and accurate acetabular segmentation by the DAE model. The following discussion explores the key findings in depth, addressing the strengths and limitations of our automated FHC/DAE measurement system.

Beyond automation, the reliability of CHD assessment is ultimately limited by the diagnostic metrics that define it. While the NA has been a longstanding reference, its methodological limitations and susceptibility to anatomical variability challenge its role as a universal standard [6,9]. These concerns warrant a closer examination of complementary approaches, particularly the FHC/DAE metric, which offers distinct advantages.

A primary limitation of the NA is its dependence on acetabular morphology, which varies considerably across breeds. Studies have shown that using a universal NA cutoff of $\geq 105°$ leads to inconsistencies and a high rate of false positives, potentially excluding genetically valuable dogs from breeding populations [6,9]. Tomlinson et al. highlighted that the prominence of the CrAR directly influences the NA, with more pronounced rims leading to larger angles [9]. Conversely, structural irregularities such as CrAR notching, characterized by indentations along the rim that disrupt its continuity, steepen the cranial curvature, and result in a flattened appearance, can lower the NA [2,9]. Additionally, osteophyte formation, which refers to localized new bone formation, on the cranial aspect of the acetabular margin can create the appearance of a double rim in the VDHE view, making the CrAR more difficult to identify [34,35]. Furthermore, variations in acetabular depth contribute to breed-dependent differences [9], introducing additional inconsistencies.

Both the NA and the FHC/DAE metrics require precise definition of the femoral head and its center. However, they differ in the other anatomical landmarks upon which they

rely. While the NA depends on the CrAR, the FHC/DAE is determined by the DAE. In this context, we believe that the FHC/DAE metric offers distinct practical advantages. The DAE, representing the outermost part of the acetabulum where the femoral head's articulation terminates dorsally, is a more geometrically consistent landmark compared to the CrAR. The DAE's greater length and continuity make it easier to identify radiographically, even in cases of moderate DJD. The DAE remains visible along most of its length, even though it appears dorsal to the femoral neck in the VDHE view, becoming difficult to define only in extreme cases of significant abnormal bone growth on the femoral head, femoral neck, or acetabulum, or in cases of severe joint deformation [34,35]. This robustness makes the DAE a more practical and reliable reference point for assessing femoral head coverage, particularly in the scope of DL models that rely on precise pixel-level detection to define anatomical structures through semantic segmentation.

Beyond the inherent anatomical variability of the NA, additional inconsistencies result from its categorization within the FCI and Flückiger's CHD scoring protocols. The classification relies on predefined NA thresholds, but overlapping criteria, such as cases where an NA of 105° or greater is classified differently depending on slight variations in joint space [3], introduce ambiguity. This lack of strict boundary distinctions, combined with subjective interpretation of anatomical features, can lead to misclassification, particularly in borderline cases. In contrast, the FHC/DAE metric offers a clear advantage by providing well-defined category boundaries, minimizing overlap and improving classification clarity.

Accurate localization of anatomical landmarks on the femoral head is crucial for the automated FHC/DAE measurement system, as the precision of the FHC depends on the exact placement of the predicted keypoints. The results indicate that the FHC detection model achieved highly precise localization, with evaluation metrics such as ED, MAE, and MSE confirming minimal deviations from ground truth annotations.

In medical imaging, segmentation plays a fundamental role in accurately delineating anatomical structures, facilitating precise identification and isolation of regions critical for measurement [36]. In this study, the DAE segmentation model demonstrated strong performance in acetabulum delineation, ensuring accurate identification of the DAE for FHC/DAE measurement calculations. The DS and IoU scores further validated the segmentation process, confirming a high degree of overlap with expert-annotated ground truth masks.

Compared to our previous acetabulum segmentation models, the current model demonstrated improved performance, benefiting from a significantly larger training dataset. In earlier studies [22,24], a DS of 0.93 and an IoU of 0.88 were achieved using a training set of only 138 images. By contrast, the current model, trained on 624 images, achieved a DS of 0.96 and an IoU of 0.92, reflecting a notable improvement in segmentation accuracy.

Beyond our prior work, this model also outperformed other AI-based radiographic segmentation models, such as those developed by Rouzrokh et al. for measuring acetabular component angles in total hip arthroplasty in humans. Their system, which focused on surgical outcomes and implant positioning, reported a DS of 0.90 for acetabular component segmentation in their anteversion angle model, the highest among their tested models [37]. The superior segmentation accuracy of our model highlights its potential to enhance diagnostic precision and streamline workflows, not only in veterinary medicine but possibly in future applications for human medicine.

In addition to its overall performance, the automated FHC/DAE system offers a substantial advantage in processing time. On a workstation equipped with an NVIDIA RTX 3090 GPU with 24 GB of VRAM, the average processing time per radiograph was 264.8 milliseconds for segmentation and 99.2 milliseconds for keypoint detection, allowing the system to generate FHC/DAE measurements for both hip joints in under 0.5 s. Al-

though performance may vary with different hardware configurations, the system remains significantly faster than manual assessment. In comparison, an experienced examiner typically requires between 1 and 1.5 min per radiograph using Dys4Vet's manual annotation tools, depending on image quality and complexity. By dramatically reducing annotation time, the automated system improves workflow efficiency and offers clear practical benefits for clinical use, particularly in large-scale screening scenarios or research environments.

To validate the FHC/DAE system, its classifications and measurements were compared with expert examiners and their agreement with each other. Agreement between E1 and E2 was almost perfect ($\kappa = 0.90$ (95% CI [0.84, 0.95], $p < 0.001$). The contingency table shows that 80.7% of classifications fell within in the "Superimposed" (67 cases) and "Lateral" (46 cases) categories. Although this distribution could suggest that the high observed agreement ($P_0$) of 90% is influenced by prevalence, the weighted kappa value and narrow confidence interval confirm that agreement extends beyond chance. Importantly, disagreements were minimal (14 cases) and primarily confined to adjacent categories such as "Superimposed" versus "Lateral", which are less penalized in the linear weighting scheme. Clinically significant categories such as "Lateral+" (10 out of 11 agreements) and "Luxation" (1 out of 1 agreement) also demonstrated strong concordance. For continuous FHC/DAE measurements, inter-rater reliability between E1 and E2 was excellent (ICC = 0.97, 95% CI [0.96, 0.98], $p < 0.001$), with a SEM of 0.91 mm, indicating that approximately 68% of E2's measurements would fall within $\pm 0.91$ mm of E1's, reinforcing the precision and consistency of the measurements.

Agreement between E1 and the system was almost perfect ($\kappa = 0.86$ 95% CI [0.80, 0.92], $p < 0.001$). The contingency table shows that 72.1% of classifications fell within the "Superimposed" (62 cases) and "Lateral" (39 cases) categories. Although this distribution reflects a common pattern in the dataset, it does not appear to overly inflate the kappa value, as demonstrated by the narrow confidence interval [28]. Importantly, disagreements were minimal (18 cases) and primarily confined to adjacent categories, limiting their clinical impact. Clinically significant categories such as "Lateral+" (9 out of 11 agreements) and "Luxation" (1 out of 1 agreement) also demonstrated strong concordance, supporting the system's ability to replicate E1's classifications across both frequent and less common categories, thus reinforcing its clinical applicability. For continuous FHC/DAE measurements, inter-rater reliability between E1 and the system was excellent (ICC = 0.97, 95% CI [0.95, 0.98], $p < 0.001$), with an SEM of 0.92 mm, indicating that approximately 68% of the system's measurements would fall within $\pm 0.92$ mm of E1's. Given that categories such as "Lateral" ($-5$ to $< -1$ mm) and "Lateral+" ($-10$ to $-6$ mm) each have a 4 mm range, this small error margin remains well within acceptable limits, reducing the likelihood of clinically significant misclassifications and supporting the system's ability to replicate expert assessments with minimal variability.

The comparison between E2 and the system showed substantial agreement ($\kappa = 0.79$, 95% CI [0.72, 0.86], $p < 0.001$), with an observed agreement ($P_0$) of 80%. Although this was slightly lower than the agreement observed with E1, it aligns with the study design, as E1 provided the ground truths used to train the system. This likely shaped the system's classification tendencies to align more closely with E1's methodology for the FHC/DAE metric. The contingency table shows that most classifications were again concentrated in the "Superimposed" (56 cases) and "Lateral" (37 cases) categories, which together accounted for 66.4% of the observations. Disagreements were more frequent than in the E1 comparison (25 cases).

Notably, the contingency tables, along with the mean $\pm$ standard deviation values of continuous FHC/DAE measurements, indicate that the system tends to assign slightly less severe classifications than both E1 and E2. The mean $\pm$ standard deviation was

$-1.32 \pm 2.70$ mm for E1, $-1.53 \pm 2.70$ mm for E2, and $-1.12 \pm 2.59$ mm for the system, reflecting a modest upward shift. This trend was more apparent in the comparison with E2, where some cases classified as "Lateral+" by E2 were assigned to "Lateral" by the system, and cases classified as "Lateral" were occasionally assigned to "Superimposed". Although these discrepancies were mostly limited to adjacent categories, which generally carry limited clinical consequence, the system's slightly conservative bias could increase the risk of false negatives in screening contexts. In particular, misclassifications near the boundary between "Superimposed" and "Lateral", which likely represents the division between normal and dysplastic hips, may result in early CHD cases being overlooked. Despite this trend, the system maintained strong agreement in less frequent but clinically important categories, with 8 out of 12 agreements for "Lateral+" and full agreement for the single "Luxation" case.

For continuous FHC/DAE measurements, inter-rater reliability between E2 and the system was excellent (ICC = 0.93, 95% CI [0.88, 0.96], $p < 0.001$), with an SEM of 1.34 mm, indicating that approximately 68% of the system's measurements would fall within $\pm 1.34$ mm of E2's. Although this SEM is higher than that observed in the E1 comparison, it remains acceptable for most categories. However, in narrower thresholds such as "Superimposed" ($-1$ to <1 mm), which spans 2 mm, and "Medial" (1 to $\leq 2$ mm), which spans just 1 mm, this level of variability could affect classifications in borderline cases.

The inclusion of a second examiner in this study helped evaluate the system's consistency with expert methodologies, ensuring conformity with established FHC/DAE classification standards. While E1's involvement in training the system understandably led to closer agreement between E1 and the system, the high agreement and reliability observed between E1 and E2 reflect consistent judgment among experienced examiners. This strong inter-rater reliability ensures that the methodologies of both examiners can be used interchangeably for FHC/DAE assessments, providing confidence that their ratings are robust and unbiased [27]. Furthermore, this consistency strengthens the validity of the system's classifications, as its performance aligns with experts whose methodologies are both reliable and consistent. Even when the system's classifications diverge slightly, particularly from those of E2, the robust examiner agreement supports the system's broader applicability.

However, it is important to note that the FHC/DAE system is not intended to function as a standalone diagnostic tool for assessing CHD. While it addresses some limitations of NA-centric scoring, it also shares certain biomechanical constraints. Like the NA, its accuracy may be affected by artificial tightening of the joint capsule in the VDHE position, which can mask subluxation and potentially lead to underestimation of CHD severity. Therefore, although FHC/DAE offers objective spatial information on the femoral head's position relative to the acetabulum, it should be interpreted alongside complementary metrics to provide a more complete picture of hip joint integrity. This integrated approach ensures a more thorough and reliable evaluation of CHD, reducing the potential impact of variability of any single metric. Looking ahead, the development of comprehensive AI-assisted diagnostic platforms that integrate multiple automated metrics may further enhance the objectivity, reproducibility, and clinical utility of CHD radiographic evaluation. The Dys4Vet software, currently under development, is one such initiative that aims to consolidate interpretable metrics, including the FHC/DAE measurement, into a unified framework for decision support. In this context, the FHC/DAE system serves as a foundational component, contributing precise and standardized data to a broader diagnostic strategy.

Importantly, these technologies are not intended to replace expert clinical judgment. While AI can assist by offering consistent and quantifiable assessments, it does not replicate the contextual understanding, ethical reasoning, or empathetic engagement that are central

to veterinary practice. Sensitivity to animal behavior, nuanced clinical interpretation, and compassionate care remain uniquely human competencies. As such, platforms like the Dys4Vet software should be viewed as tools to support and inform veterinary decision-making, not substitutes for professional expertise.

A key strength of this study is the diversity of the dataset, which includes dogs from two distinct geographical regions, Portugal and Denmark. While the specific breeds of all dogs used in model development were not fully available due to metadata limitations, the inclusion of animals from different regions inherently introduces breed variation. Additionally, the test set, for which breed data were available, consisted of a diverse range of breeds, further supporting the system's ability to generalize. Furthermore, the use of different X-ray equipment introduced natural variations in image quality and anatomical presentation, enhancing the robustness of the automated system. This variability reduces the risk of overfitting to specific acquisition parameters and reinforces its applicability across different clinical settings and animal populations. However, since breed-specific biases could not be fully assessed for the training set, further studies with fully annotated breed metadata would be beneficial for a more comprehensive evaluation of potential breed-related effects.

A key limitation of this study is that the system does not account for uncertainty in its classifications, leading it to provide definitive predictions even in cases where its confidence is low. This was particularly evident in the slight divergence observed between E2 and the system, which highlights the potential for variability in classifications, especially in borderline cases. While overall agreement remained strong, this discrepancy suggests that the model may struggle with ambiguous cases where even expert judgment varies, introducing potential inconsistencies in classification. These discrepancies appear to be more closely associated with technical image quality, such as poor contrast, blurred anatomical edges, or low image resolution. It is worth noting that the model was trained on resized, lower-resolution images, and the resulting predictions were subsequently upscaled to the original resolution. Although a smoothing filter was applied after prediction, this process may still lead to some inaccuracies. On the other hand, the FHC/DAE system demonstrated consistent performance across mild-to-severe CHD cases, suggesting that inaccuracies are not strongly influenced by disease-related anatomical variation. Without a mechanism to assess and indicate uncertainty, the system may misclassify difficult cases instead of flagging them for expert review. To address this, future improvements could incorporate uncertainty quantification methods, allowing the system to assess its confidence in each classification. By flagging low confidence cases for expert evaluation instead of making definitive predictions in uncertain situations, the system could improve reliability and reduce misclassification rates [38]. Additionally, implementing confidence-aware training strategies, where the model is trained to recognize and adapt to uncertain cases, could enhance its ability to handle borderline scenarios more effectively, further refining its classification performance [39].

## 5. Conclusions

Through the integration of segmentation and keypoint detection, the automated FHC/DAE measurement system achieved high concordance with expert evaluations and delivered consistent, time-efficient performance. Its effectiveness across both common and less frequent classification categories highlights its potential as a reliable and objective component for future integration into automated CHD scoring frameworks.

**Author Contributions:** Conceptualization, M.G. and P.F.-G.; methodology, M.G., L.G., S.A.-P. and P.F.-G.; software, L.G., V.F., P.L. and M.F.; validation, L.G., V.F., F.M. and M.F.; formal analysis, M.G., F.M. and B.C.; investigation, M.G.; resources, M.G. and F.M.; data curation, P.F.-G.; writing—original

draft preparation, P.F.-G.; writing—review and editing, P.F.-G. and M.G.; visualization, S.A.-P. and B.C.; supervision, M.G.; project administration, M.G.; funding acquisition, M.G., S.A.-P. and B.C. All authors have read and agreed to the published version of the manuscript.

**Funding:** This work was financed by the project Dys4Vet (POCI-01-0247-FEDER-046914), co-financed by the European Regional Development Fund (ERDF) through COMPETE2020, the Operational Programme for Competitiveness and Internationalisation (OPCI).

**Institutional Review Board Statement:** Ethical review and approval were waived for this study due to the observational and retrospective nature of this study.

**Informed Consent Statement:** Owner consent was waived for this study due to the observational and retrospective nature of this study.

**Data Availability Statement:** The data presented in this study are available from the corresponding author on request.

**Acknowledgments:** The authors acknowledge the UTAD Veterinary Teaching Hospital and Danish Kennel Club for allowing access to images from their data archive. The authors are also grateful for all the conditions made available by FCT—Portuguese Foundation for Science and Technology, under the projects UIDB/00772/2020, LA/P/0059/2020, https://doi.org/10.54499/UIDB/00772/2020 and Scientific Employment Stimulus—Institutional Call—CEECINST/00127/2018 UTAD, https://doi.org/10.54499/CEECINST/00127/2018/CP1501/CT0008.

**Conflicts of Interest:** Authors P.L. and M.F. were employed by Neadvance Machine Vision SA. The remaining authors declare that the research was conducted in the absence of any commercial or financial relationships that could be construed as a potential conflict of interest.

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
