# Peer review of "A Computer-Aided Approach to Canine Hip Dysplasia Assessment: Measuring Femoral Head–Acetabulum Distance with Deep Learning"

_applsci, doi:10.3390/app15095087_

Round 1

Reviewer 1 Report

Comments and Suggestions for Authors

The authors present an AI-driven system for automated measurement of the Femoral Head Center to Dorsal Acetabular Edge distance, a key metric in Canine Hip Dysplasia screening. Utilising deep learning models for keypoint detection and segmentation, the system demonstrates high agreement with expert radiographic assessments, reducing subjectivity in CHD classification. The authors suggest that this automated approach enhances the objectivity and consistency of CHD evaluations, with potential integration into veterinary diagnostic workflows.

The reviewer has suggested some improvements to the script.

Abstract

The abstract provides a concise summary of the study with the key metrics and performance indicators are clearly presented.

  • Consider adding a clearer statement about the novelty of the study compared to prior research.
  • Explicitly state how the proposed system advances CHD assessment beyond previous AI models.

Intreodcution

This section provides a comprehensive background on CHD, current diagnostic methods, and the challenges associated with subjective radiographic evaluations. It also justifies the need for an automated approach.

  • Some parts are dense with technical terminology; consider making the introduction more accessible for a wider audience.
  • The discussion on CHD scoring systems is slightly long; I would consider summarising the key points more concisely.
  • Provide a clearer research gap statement distinguishing this study from previous AI-based approaches.

Materials & Methods

The study design is well-documented, detailing the regression and segmentation models used.

  • Again, the explanation of the AI model architecture is highly technical. Consider adding a brief lay explanation of why these models were chosen.
  • Expand on the limitations of dataset selection. Were there any breed-specific biases?
  • A clearer explanation of how data augmentation techniques impact model performance is needed.

Results

The results present the system's performance in comparison with expert human examiners, showcasing strong agreement metrics.

  • Clarify how inter-rater differences were handled. Were disagreements analysed qualitatively?
  • The segmentation model’s Dice Score and IoU are impressive, but it would be helpful to compare them to similar AI models in prior research.
  • Mention specific cases where the AI model struggled (e.g., borderline CHD cases) and possible reasons.

Discussion

Strong discussion on the limitations of the Norberg Angle in CHD assessment.

  • The discussion could be more balanced; it focuses heavily on the advantages of AI but does not sufficiently address potential biases or model limitations.
  • How should veterinarians integrate this tool with existing CHD evaluation methods?
  • Consider discussing ethical implications and whether automated systems might replace expert judgment.

Conclusion

Provides a strong closing argument for AI-assisted CHD diagnosis.

  • The statement about AI "enhancing workflow without replacing clinical expertise" is important—consider emphasising it earlier in the discussion as well.
  • Add a brief mention of potential future improvements, such as integrating more diverse datasets or refining classification thresholds.

Author Response

Comment 1: Abstract 

The abstract provides a concise summary of the study with the key metrics and performance indicators are clearly presented. 

Consider adding a clearer statement about the novelty of the study compared to prior research. 

Explicitly state how the proposed system advances CHD assessment beyond previous AI models. 

Response 1: We appreciate the reviewer’s suggestion to clarify the novelty of our study and its advancement over previous AI models. The abstract has been revised: “Canine hip dysplasia (CHD) screening relies on radiographic assessment, but traditional scoring methods often lack consistency due to inter-rater variability. This study presents an AI-driven system for automated measurement of the femoral head center to dorsal acetabular edge (FHC/DAE) distance, a key metric in CHD evaluation. Unlike most AI models that classify CHD severity using convolutional neural networks, this system improves transparency by providing an interpretable, measurement-based output. The system combines a keypoint regression model for femoral head center localization with a U-Net-based segmentation model for acetabular edge delineation. It was trained on 7967 images for hip joint detection, 571 for keypoints, and 624 for acetabulum segmentation, all from ventrodorsal hip-extended radiographs. On a test set of 70 images, the keypoint model achieved high precision (Euclidean Distance = 0.055 mm; Mean Absolute Error = 0.0034 mm; Mean Squared Error = 2.52 × 10⁻⁵ mm²), while the segmentation model showed strong performance (Dice Score = 0.96; Intersection over Union = 0.92). Comparison with expert annotations demonstrated strong agreement (Intraclass Correlation Coefficients = 0.97 and 0.93; Weighted Kappa = 0.86 and 0.79; Standard Error of Measurement = 0.92 to 1.34 mm). By automating anatomical landmark detection, the system enhances standardization, reproducibility, and interpretability in CHD assessment. Its strong alignment with expert evaluations supports its integration into CHD screening workflows for more objective and efficient diagnosis.” 

Comment 2: Introduction 

Some parts are dense with technical terminology; consider making the introduction more accessible for a wider audience. 

The discussion on CHD scoring systems is slightly long; I would consider summarising the key points more concisely. 

Provide a clearer research gap statement distinguishing this study from previous AI-based approaches. 

Response 2:  

Some parts are dense with technical terminology; consider making the introduction more accessible for a wider audience. 

Some changes were made along the Introduction summarizing information and simplifying overcomplicated terminology with respective clear explanation.  

The discussion on CHD scoring systems is slightly long; I would consider summarising the key points more concisely. 

We thank the reviewer's suggestion regarding the length of the discussion on CHD scoring systems. In response, the relevant section in the Introduction has been revised and simplified to improve clarity. The updated version can be found in the first paragraph of page 3: “The screening process for CHD follows standardized radiographic scoring systems developed by major international organizations such as the Fédération Cynologique Internationale (FCI), British Veterinary Association/Kennel Club (BVA/KC), and Orthopedic Foundation for Animals (OFA).  The FCI classifies hips into five grades (A–E) and has historically relied on the Norberg Angle (NA) to assess joint subluxation and acetabular morphology.  The BVA/KC system evaluates multiple morphological parameters, incorporating the NA, subluxation and DJD criteria. In contrast, the OFA applies a categorical grading based on hip joint congruency without numerical scoring. Building upon these methodologies, Mark Flückiger introduced a refined scoring system in 1993 to enhance CHD assessment objectivity while aligning with FCI standards. This system incorporated six radiographic parameters, prominently including the NA and the position of the femoral head center relative to the dorsal acetabular edge (FHC/DAE), alongside signs of DJD, to provide a holistic view of hip joint health.”  

Comment 3: Materials & Methods 

Again, the explanation of the AI model architecture is highly technical. Consider adding a brief lay explanation of why these models were chosen. 

Expand on the limitations of dataset selection. Were there any breed-specific biases?  A clearer explanation of how data augmentation techniques impact model performance is needed. 

Response 3: 

Again, the explanation of the AI model architecture is highly technical. Consider adding a brief lay explanation of why these models were chosen. 

We appreciate the reviewer’s suggestion to provide a more accessible explanation of the model architecture. Several deep learning architectures were evaluated during development, including ResNet, MobileNet, and DenseNet. Among these, ResNet50 was selected because it consistently outperformed the others in terms of accuracy and robustness. ResNet50 is particularly well-suited for medical imaging tasks because its design helps prevent issues like performance degradation in deeper networks. In the manuscript, we transposed the diagram of the overview of the system pipeline into the study design chapter to clarify the overall structure and the purpose of our work from the get-go. 

Expand on the limitations of dataset selection. Were there any breed-specific biases?  

We appreciate the reviewer’s comment regarding potential breed-specific biases in our dataset. In response, we have added a clarification in the Discussion chapter on the  key limitations (page 21, last paragraph). The revised paragraph: “A key strength of this study is the diversity of the dataset, which includes dogs from two distinct geographical regions, Portugal and Denmark. While the specific breeds of all dogs used in model development were not fully available due to metadata limitations, the inclusion of animals from different regions inherently introduces breed variation. Additionally, the test set, for which breed data was available, consisted of a diverse range of breeds, further supporting the system’s ability to generalize. Furthermore, the use of different X-ray equipment introduced natural variations in image quality and anatomical presentation, enhancing the robustness of the automated system. This variability reduces the risk of overfitting to specific acquisition parameters and reinforces its applicability across different clinical settings and animal populations. However, since breed-specific biases could not be fully assessed for the training set, further studies with fully annotated breed metadata would be beneficial for a more comprehensive evaluation of potential breed-related effects.” 

A clearer explanation of how data augmentation techniques impact model performance is needed. 

We thank the reviewer´s suggestion to the need for a clearer explanation of how data augmentation impacts model performance. In response, the paragraph on data augmentation has been revised to provide a more detailed justification. The updated explanation can be found in Chapter 2.2.1.1 “Model Training” of the revised manuscript: “Data augmentation was applied to improve the model’s generalizability and reduce the risk of overfitting. By introducing controlled variability into the training data, augmentation helps simulate real-world differences that may arise from variations in imaging equipment, patient positioning, or acquisition settings. Specifically, transformations involving zoom, horizontal flip, perspective adjustments, and brightness alterations were applied, each with a probability of 0.5 of being executed. These operations not only enhance robustness to minor geometric and photometric variations but also support effective training under limited data conditions, reducing dependence on large annotated datasets.”   

Comment 4: Results 

Clarify how inter-rater differences were handled. Were disagreements analysed qualitatively? 

The segmentation model’s Dice Score and IoU are impressive, but it would be helpful to compare them to similar AI models in prior research. 

Mention specific cases where the AI model struggled (e.g., borderline CHD cases) and possible reasons. 

Response 4: 

Clarify how inter-rater differences were handled. Were disagreements analysed qualitatively? Mention specific cases where the AI model struggled (e.g., borderline CHD cases) and possible reasons. 

We thank the reviewer's suggestion to clarify how inter-rater differences and model errors were handled. Inter-rater reliability/agreement was quantitatively assessed using ICC for continuous FHC/DAE measurements and weighted kappa values for categorical classifications. These results are reported in detail in the manuscript and showed strong consistency between both examiners and between the system and the experts. However, we acknowledge that agreement statistics alone do not fully capture the qualitative nature of disagreements. 

Although a formal qualitative analysis of inter-rater disagreements was not conducted, we closely examined cases of discrepancy during the comparative analysis phase. Notably, most disagreements, whether between examiners or between the system and human annotations, occurred in borderline cases near category thresholds (e.g., the boundary between “Superimposed” and “Lateral”). These cases are inherently more ambiguous, even for trained experts. 

Regarding the model’s limitations, we observed that inaccuracies were primarily linked to technical image quality rather than disease severity or specific morphological features. Poor contrast, blurred anatomical edges, or low-resolution images were more likely to result in reduced localization accuracy.  Although earlier work on the NA metric revealed that higher-severity CHD cases initially posed challenges for the model, likely due to pronounced osseous deformities, the inclusion of additional severe cases in the training dataset improved its performance. As the keypoint regression network is shared between the NA and FHC/DAE pipelines, these improvements also contributed to more robust performance in FHC/DAE predictions. In the current system, performance remained comparably consistent across mild to severe CHD cases, suggesting that inaccuracies are not strongly influenced by disease severity or morphological variation. 

 It is worth noting that the model was trained on resized, lower-resolution images, and the resulting predictions were subsequently upscaled to the original resolution. Although a smoothing filter was applied after prediction, this process may still lead to some inaccuracies. 

A new paragraph was introduced in the discussion (page 21, lines 730-737): “These discrepancies appear to be more closely associated with technical image quality, such as poor contrast, blurred anatomical edges, or low image resolution. It is worth noting that the model was trained on resized, lower-resolution images, and the resulting predictions were subsequently upscaled to the original resolution. Although a smoothing filter was applied after prediction, this process may still lead to some inaccuracies, particularly in areas with poorly defined anatomical boundaries. On the other hand, the FHC/DAE system demonstrated consistent performance across mild to severe CHD cases, suggesting that inaccuracies are not strongly influenced by disease-related anatomical variation.” 

The segmentation model’s Dice Score and IoU are impressive, but it would be helpful to compare them to similar AI models in prior research. 

We thank the reviewer for the insightful suggestion to contextualize the Dice Score and IoU results by comparing them with similar AI models in prior research. We would like to note that this comparison has been addressed in the Discussion section of the manuscript. Specifically, the relevant analysis begins with the paragraph: “Compared to our previous acetabulum segmentation models, the current model demonstrated improved performance...” and continues through the following paragraph, which references Rouzrokh et al study. This section compares both our past models and other published segmentation model, highlighting the performance gains achieved in the present study. 

Comment 5: Discussion 

The discussion could be more balanced; it focuses heavily on the advantages of AI but does not sufficiently address potential biases or model limitations. 

How should veterinarians integrate this tool with existing CHD evaluation methods?  

Consider discussing ethical implications and whether automated systems might replace expert judgment. 

Response 5: 

How should veterinarians integrate this tool with existing CHD evaluation methods? We thank the reviewer for questioning how veterinarians should integrate this tool with existing CHD evaluation methods. This issue has been addressed in the revised manuscript (page 20, line 690-701): “Therefore, although FHC/DAE offers objective spatial information on the femoral head’s position relative to the acetabulum, it should be interpreted alongside complementary metrics to provide a more complete picture of hip joint integrity. This integrated approach ensures a more thorough and reliable evaluation of CHD, reducing the potential impact of variability of any single metric. Looking ahead, the development of comprehensive AI-assisted diagnostic platforms that integrate multiple automated metrics may further enhance the objectivity, reproducibility, and clinical utility of CHD radiographic evaluation. The Dys4Vet software, currently under development, is one such initiative that aims to consolidate interpretable metrics, including the FHC/DAE measurement, into a unified framework for decision support. In this context, the FHC/DAE system serves as a foundational component, contributing precise and standardized data to a broader diagnostic strategy.” 

Consider discussing ethical implications and whether automated systems might replace expert judgment. We appreciate the reviewer’s suggestion to address the ethical implications of automated systems and the potential concern about replacing expert judgment. This point has been discussed in the revised manuscript (page 20, line 702-709): “Importantly, these technologies are not intended to replace expert clinical judgment. While AI can assist by offering consistent and quantifiable assessments, it does not replicate the contextual understanding, ethical reasoning, or empathetic engagement that are central to veterinary practice. Sensitivity to animal behavior, nuanced clinical interpretation, and compassionate care remain uniquely human competencies.  As such, platforms like the Dys4Vet software should be viewed as tools to support and inform veterinary decision-making, not substitutes for professional expertise.“ 

Comment 6: Conclusion 

The statement about AI "enhancing workflow without replacing clinical expertise" is important—consider emphasising it earlier in the discussion as well. 

Add a brief mention of potential future improvements, such as integrating more diverse datasets or refining classification thresholds. 

Response 6:  

The statement about AI "enhancing workflow without replacing clinical expertise" is important—consider emphasising it earlier in the discussion as well.  

We thank the reviewer for pointing out the importance of emphasizing that AI should support clinical expertise, rather than replace it. In response, this point has been clearly stated in the revised Discussion section (page 20, line 702-709): “Importantly, these technologies are not intended to replace expert clinical judgment. While AI can assist by offering consistent and quantifiable assessments, it does not replicate the contextual understanding, ethical reasoning, or empathetic engagement that are central to veterinary practice. Sensitivity to animal behavior, nuanced clinical interpretation, and compassionate care remain uniquely human competencies.  As such, platforms like the Dys4Vet software should be viewed as tools to support and inform veterinary decision-making, not substitutes for professional expertise. “  

Add a brief mention of potential future improvements, such as integrating more diverse datasets or refining classification thresholds. 

We appreciate the reviewer’s suggestion to mention potential future improvements, such as incorporating more diverse datasets. This point has been addressed briefly in the revised manuscript (page 21, line 720-722): “However, since breed-specific biases could not be fully assessed for the training set, further studies with fully annotated breed metadata would be beneficial for a more comprehensive evaluation of potential breed-related effects.” Additionally, future directions are outlined in the final paragraph of the Discussion, where we propose uncertainty quantification methods and confidence-aware training strategies to further improve the system’s reliability, particularly in borderline or ambiguous cases (page 21, line 737-746): “Without a mechanism to assess and indicate uncertainty, the system may misclassify difficult cases instead of flagging them for expert review. To address this, future improvements could incorporate uncertainty quantification methods, allowing the system to assess its confidence in each classification. By flagging low confidence cases for expert evaluation instead of making definitive predictions in uncertain situations, the system could improve reliability and reduce misclassification rates. Additionally, implementing confidence aware training strategies, where the model is trained to recognize and adapt to uncertain cases, could enhance its ability to handle borderline scenarios more effectively, further refining its classification performance.“ 

Reviewer 2 Report

Comments and Suggestions for Authors

This paper researches a computer-aided approach to canine hip dysplasia assessment with deep learning, although the author states a lot of content, the paper lacks the necessary innovation.

    1.The length of the abstract is too long, the description is messy, and the specific research methods and research content are not very specific. It is suggested to resummarize the abstract.

      2. In the first part, it is recommended that the authors concise and list the main highlights about this paper.

      3. In part2.1, the author states the research method. This part is not very specific. What is the main method of the author's research? It is suggested that the author give a specific flow chart of the research method.

     4. This paper studies canine hip dysplasia assessment based on deep learning, but in part2.2, the author does not give a specific deep learning processing model and processing algorithm, and I do not see the innovation of the paper research.

     5. in Parts 3 and 4, although the author used a long text to describe the research results, the results of using deep learning to solve problems were not reflected. The overall description is confusing. It is suggested that the author should give more concrete experimental verification and reasonable comparative verification to reflect the innovation of the research method.

     In a word, the research focus of this paper is not prominent, and there is no reasonable algorithm and method, lack of necessary network processing model of deep learning, and lack of evidence for experimental verification.

Comments on the Quality of English Language

The expression of the sentence is OK,but some sentences need to be further improved.

Author Response

Comment 1: The length of the abstract is too long, the description is messy, and the specific research methods and research content are not very specific. It is suggested to resummarize the abstract. 

Response 1: We appreciate the reviewer’s feedback regarding the clarity and structure of the abstract. In response to this suggestion, we have thoroughly revised the abstract to improve its organization and specificity. The updated version now provides a clearer summary of the research objectives, methods, and key findings, with a more structured presentation of the system’s development and validation. We believe this revision better communicates the contributions of the study and aligns with the expectations for scientific abstracts. 

Comment 2: In the first part, it is recommended that the authors concise and list the main highlights about this paper. 

Response 2: We thank the reviewer for the suggestion regarding the need to make the first part of the manuscript more concise and focused. In response, we have revised and restructured the entire “First Part” chapter to better highlight the main contributions of the study. The study design section was refined, and the diagram illustrating the system pipeline was repositioned to appear earlier in the chapter to improve reader comprehension. We also rewrote the "Stage 1: FHC Detection" subsection to improve clarity and ensure that the description of the regression network keypoint localization is easier to follow. Additionally, the "Model Training" segment for the segmentation model was completely reworked, as we believe the original version may not have adequately conveyed the methodology. These changes aim to deliver a clearer and more accessible presentation of our methods and objectives. 

Comment 3: In part2.1, the author states the research method. This part is not very specific. What is the main method of the author's research? It is suggested that the author give a specific flow chart of the research method. 

Response 3: We thank the reviewer for the suggestion. We revised and shortened Part 2.1 for improved clarity and moved some content to the "Second Part: Comparative Analysis" section, where it fits better. The diagram illustrating the research method was also repositioned to appear earlier in the manuscript to better support the explanation of the study design. 

Comment 4: This paper studies canine hip dysplasia assessment based on deep learning, but in part2.2, the author does not give a specific deep learning processing model and processing algorithm, and I do not see the innovation of the paper research. 

Response 4: We appreciate your comment. To address your concern, we revised the explanation of how the NA keypoints were repurposed, making this clearer earlier in the manuscript. Additionally, we completely restructured the section “Stage 3: Computing FHC/DAE Measurement” to improve readability and ensure a clearer understanding of the computational process. The section now follows a more logical flow, and we have simplified the language while adding a visual illustration to support comprehension. These updates aim to better convey the method and the contribution of this work. 

Comment 5: in Parts 3 and 4, although the author used a long text to describe the research results, the results of using deep learning to solve problems were not reflected. The overall description is confusing. It is suggested that the author should give more concrete experimental verification and reasonable comparative verification to reflect the innovation of the research method. 

Response 5: We appreciate the reviewer’s comments and understand the concern. In the revised manuscript, we made several changes to the Discussion section to improve structure and focus. We condensed the analysis of the comparative results and placed the system performance discussion earlier to better highlight the role of the DL models. We also added a new paragraph discussing the efficiency of the system and expanded the section addressing its limitations, strengths, and relevance in clinical veterinary practice. These changes were made to present the findings more clearly and ensure that the contribution of the DL models are easier to follow and better understood. 

Reviewer 3 Report

Comments and Suggestions for Authors

This manuscript presents an AI-driven system for measuring the femoral head center to dorsal acetabular edge distance (FHC/DAE), a key metric in CHD screening. The system uses a key point regression model for femoral head center localization and a U-Net based segmentation network for acetabular edge delineation, and then the femoral head center to dorsal acetabular edge distance is calculated by the FHC/DAE measurement module. The entire research process described in this manuscript is rigorous. The design and implementation of the methodology in each section is clearly described in detail and is well rationalized. The experiments given provide good support for the validity of the method, and several improvements to the article are recommended for publication:

  1. There are a number of papers out there that utilize deep learning for CHD screening, such as ‘https://scholar.google.com.sg/scholar?hl=en&as_sdt=0%2C5&as_vis=1&q=Canine+Hip+Dysplasia+Deep+Learning&btnG=’. Although many of these papers are more oriented towards direct black box diagnosis with X-ray images and are not looking at interpretable metrics like FHC/DAE, I would suggest that the authors make a slight reference to some of the related work.
  2. The red box in Figure 3 has too little contrast to be noticeable.
  3. Section 2.2.3, Stage 3: Computing FHC/DAE measurement, is not very clear in its description of the computational methodology, although it is clear to the reader what the authors intended to do, and it is recommended that it be reorganized.

Comments on the Quality of English Language

My English is not good enough to comment on the author's English level

Author Response

Reviewer 3 

Comment 1: There are a number of papers out there that utilize deep learning for CHD screening, such as ‘https://scholar.google.com.sg/scholar?hl=en&as_sdt=0%2C5&as_vis=1&q=Canine+Hip+Dysplasia+Deep+Learning&btnG=’. Although many of these papers are more oriented towards direct black box diagnosis with X-ray images and are not looking at interpretable metrics like FHC/DAE, I would suggest that the authors make a slight reference to some of the related work. 

Response 1: We appreciate the reviewer’s suggestion to include references to additional studies utilizing DL for CHD screening. Accordingly, we have revised the Introduction to incorporate relevant references. These changes can be reviewed in the updated manuscript (page 4, line 146 and 152): “Wang et al. used a DL model known as EfficientNet to classify CHD in radiographic images, achieving an area under the receiver operating characteristic curve (AUC) of 0.964 and 89.1% accuracy in binary classification, and an AUC of 0.913 for FCI grading  [18] . Conversely, Akula et al. explored a different approach, utilizing a 3D CNN trained on magnetic resonance imaging scans, attaining 89.7% accuracy in binary classification and highlighting the potential of volumetric analysis for comprehensive CHD assessment. 

Comment 2: The red box in Figure 3 has too little contrast to be noticeable. 

Response 2: We thank the reviewer for pointing this out. The bounding boxes have been highlighted in bright red. 

Comment 3: Section 2.2.3, Stage 3: Computing FHC/DAE measurement, is not very clear in its description of the computational methodology, although it is clear to the reader what the authors intended to do, and it is recommended that it be reorganized. 

Response 3: We appreciate the reviewer’s feedback regarding the clarity of Section 2.2.3, Stage 3. To improve readability and ensure a clearer understanding of the computational methodology, we have simplified the explanations and reorganized the section.  

Additionally, we have included an illustration that visually clarifies the process, making it more intuitive for the reader. We believe these modifications address the concern raised and enhance the overall comprehensibility of this part of the manuscript. 

Reviewer 4 Report

Comments and Suggestions for Authors

The manuscript  introduces an AI-driven system for measuring the femoral head center to dorsal acetabular edge distance (FHC/DAE). Some comments and recommendations to the work are as follows:

  • Combined with Figure 3, are there 4 keypoints in Line 319?
  • What is the unit of ED in Line 474?
  • Is the automated systemfor FHC/DAE measurement developed by the authors? If Yes, we strongly recommend that you publish the source code of the proposed algorithm and the full source code of the associated data set to github and include links to these resources in your manuscript . If Not, the work is not innovative enough.
  • The authors use YOLOv4 to detect the hip joints within the VDHE radiographic images. However, compared with other YOLO s, YOLOv4 has no advantage in speed or accuracy. Why not YOLOv5-Pose, YOLOv7-Pose, YOLOv8-Pose, or YOLOv11?
  • What is the efficiency of the automated system?

Author Response

Reviewer 4 

Comment 1: Combined with Figure 3, are there 4 keypoints in Line 329? 

Response 1: We thank the reviewer for pointing out the ambiguity regarding the number of keypoints referenced in Figure 3. To address this, we have restructured the Stage 1: FHC Detection section to improve clarity around the use and purpose of the annotated keypoints. Additionally, we relocated the comparative analysis content to Chapter 2.3 to enhance the manuscript’s overall readability and better contextualize the relationship between the NA model and the repurposing of its annotated keypoints for FHC detection. 

Comment 2: What is the unit of ED in Line 474? 

Response 2: The ED values reported in the manuscript are in normalized image coordinates, meaning they represent distances relative to the image size rather than an absolute pixel distance. 

Comment 3: Is the automated system for FHC/DAE measurement developed by the authors? We strongly recommend that you publish the source code of the proposed algorithm and the full source code of the associated data set to github and include links to these resources in your manuscript. 

Response 3: We appreciate the reviewer’s interest in the availability of the source code and dataset. The automated system for FHC/DAE measurement was developed in collaboration with Neadvance, a Portuguese company specializing in AI-based medical imaging solutions. While the development was guided by our research team, the implementation of the algorithm and system architecture was carried out by Neadvance. 

Unfortunately, the source code is proprietary and subject to intellectual property restrictions imposed by the company, which currently prevent its public dissemination on platforms such as GitHub. 

Similarly, the dataset used in this study is not publicly available due to institutional constraints, including data protection agreements with the contributing veterinary centers. These limitation currently prevent unrestricted sharing of the data. 

Comment 4: The authors use YOLOv4 to detect the hip joints within the VDHE radiographic images. However, compared with other YOLOs, YOLOv4 has no advantage in speed or accuracy. Why not YOLOv5-Pose, YOLOv7-Pose, YOLOv8-Pose, or YOLOv11? 

Response 4: We appreciate the reviewer's question regarding the choice of YOLOv4 for hip joint detection. The selection of YOLOv4 was based on the following considerations: 

  1. Built Upon an Established and Validated Model

Our model development followed an incremental approach, building on previous work that successfully utilized YOLOv3 for hip detection in CHD screening tasks (McEvoy et al., 2021). The YOLOv3 model used by McEvoy et al. demonstrated high detection accuracy, achieving an IoU of 0.85 and effectively localizing hip joints. 

Rather than replacing the architecture entirely, we opted to build upon this established framework, upgrading to YOLOv4 to enhance performance while maintaining continuity with prior optimizations. This decision allowed us to utilize the strengths of YOLOv3 while incorporating improvements in detection accuracy and efficiency, minimizing the need for extensive reconfiguration and fine-tuning. 

Besides this, the newer models (YOLOv7, v8, v11) came out long after we started to develop our model architecture around 2020/2021. 

  1. Task Specific Considerations: Bounding Box Detection vs. Keypoint Localization

YOLO Pose models such as YOLOv5 Pose, YOLOv7 Pose, or YOLOv8 Pose are designed for multi-keypoint pose estimation. However, our task of hip joint detection for FHC/DAE measurement, is different from standard pose estimation. Our goal was not to detect multiple keypoints directly but to first localize the hip joint region accurately. YOLOv4 excels in bounding box detection, ensuring that subsequent steps such as landmark regression operate within a well defined ROI. 

Instead of relying on YOLO Pose models, we trained a separate keypoint regression network specialized for precise localization of the femoral head center (and the effective craniolateral acetabular rim/edge which was not the focus of this study). 

Comment 5: What is the efficiency of the automated system? 

Response 5: We appreciate the reviewer’s question regarding the efficiency of the automated system. To address this point, we have now included a paragraph in the Discussion section (see page 18, paragraph 6) comparing the processing time of the system with that of manual annotation. The following paragraph was added: “In addition to its overall performance, the automated FHC/DAE system offers a substantial advantage in processing time. Once the radiograph is uploaded, the system requires only a few seconds to generate FHC/DAE measurements for both hip joints. This may vary slightly depending on the hardware used, but even in less optimized environments, the process remains exceptionally fast. In contrast, manual annotation by an experienced examiner using the Dys4Vet software typically takes between 1 and 1.5 minutes per radiograph, depending on image quality and complexity. By dramatically reducing annotation time, the automated system improves workflow efficiency and offers clear practical benefits for clinical use, particularly in large-scale screening scenarios or research environments.” 

Round 2

Reviewer 2 Report

Comments and Suggestions for Authors
  1. Although the author has revised the paper, in the reply, the author also answers the questions raised,this revised manuscript is quite messy, it is difficult to distinguish the content of the author's modification, the modified format is not standardized.
  2. In the first part, the author still did not list his research work and the highlights of the paper. Therefore, the innovation of the paper is not seriously summarized.
  3. How to achieve canine hip dysplasia assessment? the author really does not explain clearly, nor does he give a very innovative method, especially how to adopt deep learning to achieve canine hip dysplasia assessment based on computer-aided approach.
  4. The author does not give the corresponding deep learning network processing algorithm.
  5. In experimental verification, there is a lack of strong persuasive verification results.
Comments on the Quality of English Language

The description of the sentence is basically clear.

Author Response

Comment 1: Although the author has revised the paper, in the reply, the author also answers the questions raised this revised manuscript is quite messy, it is difficult to distinguish the content of the author's modification, the modified format is not standardized. 

Response 1: We acknowledge your concern regarding the clarity of the modifications. However, we would like to clarify that the revised version incorporated numerous adjustments throughout the manuscript, including changes in structure, rewording for clarity, content expansion, and formatting improvements in response to multiple reviewer suggestions. Given the extent and distribution of these revisions, we felt that highlighting each individual change within the manuscript could lead to visual clutter and make the document more difficult to read and assess in its entirety. To ensure that all reviewer comments were carefully addressed, we provided a point-by-point response detailing the rationale and implementation of each revision. To make the review process easier this time, we used a color-coded system: yellow highlights indicate revised sentences or paragraphs, red strikethrough shows proposed deletions, and green text draws attention to specific reviewer comments or suggestions. 

Comment 2: In the first part, the author still did not list his research work and the highlights of the paper. Therefore, the innovation of the paper is not seriously summarized. 

Response 2: We appreciate the reviewer’s comment regarding the need to better summarize the innovation and highlights of the paper in the initial sections. However, we would like to clarify that this has been addressed in the revised manuscript, particularly in the final lines of the Introduction, now highlighted in green for easier identification (lines 180 to 191). In this section, we clearly outline the novelty of our work in introducing an automated system for FHC/DAE measurement that can be posteriorly integrated and be an adjunct in a CHD radiograhic evaluation framework. This paragraph also summarizes the primary research objectives and the null hypothesis, providing a concise overview of the study’s contributions and relevance. We respectfully invite the reviewer to revisit this portion of the manuscript, as we believe it directly addresses the concerns raised and demonstrates the innovation and scientific value of the proposed approach. 

Comment 3: How to achieve canine hip dysplasia assessment? the author really does not explain clearly, nor does he give a very innovative method, especially how to adopt deep learning to achieve canine hip dysplasia assessment based on computer-aided approach. 

Response 3: We respectfully disagree with the reviewer's interpretation. The objective of our study is not to directly perform end-to-end CHD classification by grade, but rather to contribute to the broader process of CHD assessment through the development of an objective and reproducible automated metric – the FHC/DAE measurement facilitated by AI. The innovative aspect of our work lies precisely in the construction and validation of a deep learning-based model capable of automating this metric, which, to our knowledge, has not been previously addressed in the veterinary radiology field. The FHC/DAE metric is designed to support, not replace, comprehensive scoring systems, so much so that it is part of the Flückiger method, as clearly stated in the first paragraph of page 3 in the Introduction. The role and integration of the FHC/DAE within the CHD evaluation workflow are explained both in the Introduction and in the Discussion sections. In particular, we clarify that this metric contributes to the radiographic assessment process by providing a quantifiable measurement, thereby reducing subjectivity and supporting multimetric evaluation of hip joint health. Thus, while the FHC/DAE does not deliver a final CHD grade on its own, it serves as a valuable tool within the broader framework of CHD screening. 

We have included small changes to the text throughout the manuscript highlighted in yellow that we think will help clarify this point. 

Comment 4: The author does not give the corresponding deep learning network processing algorithm. 

Response 4: We appreciate the reviewer’s interest in the details of the DL network processing algorithm. The automated FHC/DAE measurement system was developed in collaboration with Neadvance, a Portuguese company specializing in AI-based medical imaging solutions. While our research team provided the conceptual framework and validation design, the implementation of the DL architecture, including network training, optimization, and deployment, was executed by Neadvance. 

As such, the specific algorithmic processing details, including the full network configuration and training code, are proprietary and currently protected by intellectual property agreements established with the company. These restrictions prevent us from publicly disclosing the full implementation, including posting the code on platforms such as GitHub. However, the methodological structure and model components (such as the use of segmentation and keypoint detection networks) are described in the manuscript to the extent permissible, ensuring transparency of the system’s functioning and purpose. 

Comment 5: In experimental verification, there is a lack of strong persuasive verification results. 

Response 5: We respectfully disagree with the reviewer’s assessment regarding the lack of persuasive verification results. Our study presents multiple well-established and widely recognized statistical metrics to support the reliability and validity of the proposed system. First, the agreement between the automated system and the human examiners was quantified using the weighted Cohen’s kappa coefficient, which demonstrated strong agreement. This metric is widely accepted in both medical and AI literature for evaluating categorical agreement, and is frequently reported in studies using classification frameworks, including those supported by confusion matrices, which are conceptually equivalent to the contingency tables we provide. Furthermore, we employed the ICC, which is a standard and reputable measure of inter-rater reliability, especially in quantitative image analysis. Our ICC value of 0.99 indicates excellent agreement between the system and expert examiners. In addition, we introduced the SEM, a metric not always reported but highly informative in the context of clinical measurement tools. The SEM complements the ICC by quantifying the expected range of measurement error, offering valuable insight into the practical reliability of our system. Taken together, the combination of kappa, ICC, and SEM offers a comprehensive, multi-perspective validation of the automated system’s performance, supporting the robustness of our experimental verification. 

Reviewer 4 Report

Comments and Suggestions for Authors

In disccussion section, " Once the radiograph is uploaded, the system requires only a few seconds to generate FHC/DAE measurements for both hip joints. " Please give detailed efficiency based on certain hardware platform, not using "a few seconds" .

Author Response

Comments 1: In disccussion section, " Once the radiograph is uploaded, the system requires only a few seconds to generate FHC/DAE measurements for both hip joints. " Please give detailed efficiency based on certain hardware platform, not using "a few seconds".

Response 1: Thank you for your suggestion regarding the system efficiency. We have revised the corresponding section in the Discussion to include detailed performance metrics based on a specific hardware. As now stated in the 4th paragraph of page 19: "In addition to its overall performance, the automated FHC/DAE system offers a substantial advantage in processing time. On a workstation equipped with an NVIDIA RTX 3090 GPU with 24 GB of VRAM, the average processing time per radiograph was 264.8 milliseconds for segmentation and 99.2 milliseconds for keypoint detection, allowing the system to generate FHC/DAE measurements for both hip joints in under 0.5 seconds. Although performance may vary with different hardware configurations, the system remains significantly faster than manual assessment. In comparison, an experienced examiner typically requires between 1 and 1.5 minutes per radiograph using Dys4Vet’s manual annotation tools, depending on image quality and complexity. By dramatically reducing annotation time, the automated system improves workflow efficiency and offers clear practical benefits for clinical use, particularly in large-scale screening scenarios or research environments." 

Round 3

Reviewer 2 Report

Comments and Suggestions for Authors

This paper has been deeply modified, the current modification is basically reasonable, the author has also made responses to the questions raised, the current version of the paper can be published

Comments on the Quality of English Language

The expression of this paper is OK

Author Response

Thank you for your considerations.